# Risk-Stratified Therapy for Pediatric Acute Myeloid Leukemia

**DOI:** 10.3390/cancers15164171

**Published:** 2023-08-18

**Authors:** Daisuke Tomizawa, Shin-Ichi Tsujimoto

**Affiliations:** 1Division of Leukemia and Lymphoma, Children’s Cancer Center, National Center for Child Health and Development, Tokyo 157-8535, Japan; 2Department of Pediatrics, Yokohama City University Graduate School of Medicine, Yokohama 236-0004, Japan; shnch@yokohama-cu.ac.jp

**Keywords:** acute myeloid leukemia, children, cytogenetics, molecular genetics, measurable residual disease, chemotherapy, hematopoietic stem cell transplantation, novel therapy

## Abstract

**Simple Summary:**

Owing to the 40-year worldwide efforts for improving diagnosis and therapy for acute myeloid leukemia (AML), the second most common type of leukemia in children, overall survival rates of children with AML have now reached 70% to 80% in developed countries. This review article comprehensively describes the history and advances in the current state-of-the-art risk-stratified therapy for AML in children. However, it is likely that the traditional approaches have already reached their limits, and therefore, novel approaches are absolutely essential. The current state and future directions for incorporating novel molecular-targeted drugs into contemporary therapy through international collaboration are also extensively discussed. These aspects present key solutions for further improvements in outcomes of children with AML.

**Abstract:**

Acute Myeloid Leukemia (AML) is the second most common type of leukemia in children. Recent advances in high-resolution genomic profiling techniques have uncovered the mutational landscape of pediatric AML as distinct from adult AML. Overall survival rates of children with AML have dramatically improved in the past 40 years, currently reaching 70% to 80% in developed countries. This was accomplished by the intensification of conventional chemotherapy, improvement in risk stratification using leukemia-specific cytogenetics/molecular genetics and measurable residual disease, appropriate use of allogeneic hematopoietic stem cell transplantation, and improvement in supportive care. However, the principle therapeutic approach for pediatric AML has not changed substantially for decades and improvement in event-free survival is rather modest. Further refinements in risk stratification and the introduction of emerging novel therapies to contemporary therapy, through international collaboration, would be key solutions for further improvements in outcomes.

## 1. Introduction

Acute Myeloid Leukemia (AML) is a form of hematopoietic malignancy characterized by clonal proliferation of immature myeloid cells. As can be seen from the fact that the median diagnostic age of AML is over 60 years old, AML is the most common type of leukemia in adults, whereas it is the second most common leukemia subtype in children, accounting for 20–25% of pediatric leukemia cases, with an incidence of approximately seven cases per 1,000,000 children per year [1]. There are no sex differences in prevalence of AML in children. Although AML in children may arise from certain constitutional chromosomal abnormalities (e.g., Down syndrome [trisomy 21]), familial predisposition syndromes or inherited gene mutations/translocations (e.g., inherited bone marrow failure syndromes), acquired conditions (e.g., myelodysplastic syndrome [MDS]), or exposure to chemotherapy/radiotherapy (therapy-related myeloid neoplasms), most of the children develop AML as a de novo disease without apparent etiology. A multi-step process of an accumulation of chromosomal and genomic alterations within immature myeloid cells results in the development of AML. Recently, novel AML classifications (Fifth edition of the World Health Organization [WHO] Classification of Haematolymphoid Tumours and International Consensus Classification [ICC] of Myeloid Neoplasms and Acute Leukemias) have been proposed [2,3,4]. Despite some existing differences, both classifications place more emphasis on molecular/genetic criteria compared to the previous ones. However, one should note that recent evidence suggests that AML in children and that in adults are distinct at least in terms of mutational landscape [5]. 

Overall Survival (OS) rates of children with AML have dramatically improved in the past 40 years, currently reaching 70–80% in developed countries [6,7,8,9,10,11,12,13,14,15,16,17]. This was accomplished mainly by the intensification of conventional chemotherapy, improvement in risk stratification, appropriate use of allogeneic Hematopoietic Stem Cell Transplantation (HSCT), and improvement in supportive care. However, despite the cytogenetic/mutational heterogeneity of the disease, the principle treatment for pediatric AML has not changed substantially for decades and improvement in Event-Free Survival (EFS) is rather modest [18]. In this review, the state-of-the-art risk-stratified therapy for children with AML other than Acute Promyelocytic Leukemia (APL) and Myeloid Leukemia associated with Down Syndrome (ML-DS) will be highlighted, including the historical background and future perspectives emphasizing risk stratification and molecularly targeted therapies.

## 2. Prognostic Factors and Risk Stratification in Pediatric AML

Risk stratification is one of the key elements for successful treatment in AML, and its aim is (A) to assign patients to therapies with sufficient intensity, (B) to avoid excess toxicities by avoiding therapies with unnecessary intensity, and recently, (C) to identify targetable lesions to incorporate targeted therapies. To properly risk-stratify patients, it is necessary to predict the treatment failure risk of patients by evaluating various prognostic factors. Prognostic factors can be subdivided into patient-associated factors (e.g., age, ethnicity) and disease-related factors (e.g., leukemia-specific cytogenetics/molecular genetics, drug resistance). Age at diagnosis is prognostic, i.e., the survival rate of children is significantly better than young adults, and that of young adults is better than older adults. However, within children < 15 years old, the impact of age difference is not significant [19]. Regarding ethnicities, analysis of Children’s Oncology Group (COG) studies mainly involving North America showed that Hispanic and African-American children had significantly worse OS rates compared to Caucasian children, and that access to chemotherapy, differences in supportive care, leukemic phenotype, and reduced compliance were unlikely to be the explanations [20]. There is no data that directly analyzed differences between Asian children and other ethnicities; however, the literature shows a higher prevalence of t(8;21) (*RUNX1*::*RUNX1T1*)-positive AML in Asian populations (approximately 30% compared to 12–14% of the U.S. or European patients) [1,21,22]. Overall, the impact of patient-associated factors is not as large as the disease-related factors in children with AML.

### 2.1. Leukemia-Specific Cytogenetics/Molecular Genetics

Leukemia-associated genetic profiles of 369 patients in the Japanese Pediatric Leukemia/Lymphoma Study Group (JPLSG) trial AML-05 are listed in Figure 1 [22]. The distribution of genetic profiles is similar to the other groups in the U.S. or Europe except that the proportion of patients with *RUNX1*::*RUNX1T1* is high in the Japanese cohort, as mentioned previously [1]. 

Analyses of clinical trials conducted from the late 1980s to the early 2000s revealed the prognostic significance of recurrent chromosomal aberrations in AML. In the United Kingdom (UK) studies MRC-AML10 and MRC-AML12, children with t(8;21)(q22;q22) (*RUNX1*::*RUNX1T1*) and inv(16)(p13q22) (*CBFB*::*MYH11*), the core-binding factor (CBF)-AML, had the best prognosis (80% OS rate), and the patients with chromosome 12 or 5q abnormalities, t(6;9)(p23;q34) (*DEK*::*NUP214*), monosomy 7, and t(9;22)(q34;q11) (*BCR*::*ABL1*) had the worst prognosis (36% OS rate) [23]. However, the majority (nearly 70%) of the patients were classified as intermediate risk (56% OS rate), which includes patients with normal karyotypes, chromosome 11q23 abnormalities, Acute Megakaryoblastic Leukemia (AMKL), and others. International collaborative efforts have significantly contributed to further uncovering the prognoses of certain AML subtypes. One of the first successes was the retrospective analysis of chromosome 11q23 abnormalities by the International BFM study group (I-BFM) consisting of 11 cooperative study groups in 15 countries [24]. 11q23 abnormalities or *KMT2A* gene rearrangements (*KMT2A*-r) account for 15–20% of pediatric AML, and recent studies have identified more than 100 fusion gene partners [25]. The 756 patients included in this study showed an “intermediate” prognosis of 44% EFS and 56% OS rates, but large EFS/OS differences were identified among the following different translocation partners: t(1;11)(q21;q23) (*KMT2A*::*MLLT11*) showed the best prognosis and t(6;11)(q27;q23) (*KMT2A*::*AFDN*), t(10;11)(p12;q23) (*KMT2A*::*MLLT10*), and t(10;11)(p11.2;q23) (*KMT2A*::*ABI1*) showed unfavorable prognoses. The other following I-BFM projects, such as t(8;16)(p11;p13)/*CREBBP*::*KAT6A* (intermediate OS, spontaneous remission in neonatal cases), t(6;9)(p23;q34)/*DEK*::*NUP214* (high-risk of relapse, improved EFS by HSCT), *KIT* and *RAS* mutations in t(8;21) (not associated with a worse outcome), t(16;21)(p11;q22)/*FUS*::*ERG* (extremely poor prognosis), t(16;21)(q24;q22)/*RUNX1*::*CBFA2T3* (favorable outcome), and hypodiploidy (poor prognosis), have also elucidated the clinical features and prognosis of these relatively rare subsets [26,27,28,29,30].

Finally, advances in molecular/genetic analyses have revealed many of the prognostic genetic markers in pediatric AML. Among non-DS AMKL (rare in adults, but 4–15% in children), inv(16)(p13q24)/*CBFA2T3*::*GLIS2* (18.4%), *KMT2A*-r (17.2%), and t(11;12)(p15;p13)/*NUP98*::*KDM5A* (11.5%) formed a poor prognostic subgroup, and mutations of the *GATA1* gene that generate the short form of GATA1 (GATA1s; 9.2%) and t(1;22)(p13;q13)/*RBM15*::*MRTFA* (10.2%) formed a good prognostic group [31,32,33]. Many of the important genetic prognostic markers were identified among the cytogenetically “normal” AML (CN-AML; approximately 40% in adult AML and 20% in pediatric AML) as well. As a poor prognostic marker, internal tandem duplication (ITD) of the *FLT3* gene (*FLT3*-ITD) is found in approximately 10% of pediatric AML and 20–30% of adult AML and is also important as a targetable marker [34]. As favorable prognostic markers, a mutation of the *NPM1* gene that generates cytoplasmic NPM1 (NPM1c) is found in approximately 5–8% of pediatric AML and 20–30% in adult AML, and biallelic *CEBPA* mutations in approximately 5% of both pediatric and adult AML [35,36]. Recent studies have revealed that *CEBPA*-basic leucine zipper (*CEBPA*-bZip) mutations are associated with favorable clinical outcomes regardless of monoallelic or biallelic mutational status (80% of the patients have a double mutation) [37]. Mutations in *IDH1*, *IDH2*, and *DNMT3A* are found in 7–14%, 8–19%, and 18–22%, respectively, of adult AML (most frequently seen in CN-AML) [38]. The prognostic significance of these mutations are not fully established, but all these mutations are extremely rare in children [39,40]. Many of the newly discovered gene mutations are less frequent in children. In addition to the already mentioned gene mutations, the *TP53* mutation is found in approximately 8% of adult AML, is associated with older age, has complex and monosomal karyotypes, has a very poor outcome, and has been given a strong emphasis in recently proposed European LeukemiaNet (ELN) 2022 recommendations, but it is rarely seen in pediatric AML [41,42]. However, *NUP98*::*NSD1* encoded by cryptic t(5;11)(q35;p15.5) was discovered in 16.1% of pediatric CN-AML, whereas it was only 2% of CN-AML in adults [43]. This fusion is associated with high leukocyte count, monocytic leukemia (M4 or M5 in French-American-British [FAB] classification), *FLT3*-ITD, and a very poor prognosis. A recent study analyzing the COG trial cohorts showed poor outcomes of not only *NUP98*::*NSD1* and *NUP98*::*KDM5A* cases but of cases with other *NUP98*-fusion-positive AML (unlike *NSD1* and *KDM5A* cases, other *NUP98*-fusions are typically not cryptic) [44]. Tandem duplication in the *UBTF* gene (*UBTF*-TD) is another example of a mutation that is predominant in pediatric AML (approximately 4% of newly diagnosed and 9% of relapsed pediatric AML). This mutation is associated with normal karyotype or trisomy 8 with co-occurring *WT1* mutations or *FLT3*-ITD and confers an unfavorable prognosis [45]. International cooperation will become increasingly important to further identify a subgroup of pediatric AML with prognostic impact, which generally would include small numbers of patients. 

### 2.2. Treatment Response Including Measurable Residual Disease 

Assessment of treatment response is regarded as an in vivo method to measure leukemia drug resistance and is widely used to risk-stratify patients with AML. In the UK MRC-AML10 study, bone marrow morphological response after initial induction therapy was significantly associated with both OS and relapse rates, and ≥15% bone marrow blasts after initial induction without favorable genetic abnormalities were allocated to the poor-risk arm in the MRC-AML12 study [46,47]. In the German BFM studies, analysis of AML-BFM83 and AML-BFM87 studies showed that residual bone marrow blasts (≥5%) at day 15 of initial induction were associated with reduced EFS rates and was therefore included in the risk group definition since the AML-BFM93 study [48]. A similar approach to risk stratification using morphological bone marrow response was used in other pediatric AML studies as well after the 1990s. 

Although >85% of children with AML achieve morphological remission after one or two courses of induction therapy, 30–40% of the patients eventually experience overt relapse. A growing need for more accurate methods to assess treatment response led to the development of molecular or immunophenotypic determination of Measurable Residual Disease (MRD) in the late 1990s to early 2000s. Regarding molecular MRD assessment, a Polymerase Chain Reaction (PCR) approach, and recently a next-generation sequencing (NGS) approach, targeting certain AML-specific genetic markers (fusion transcripts or gene mutations) can be taken with a sensitivity of 0.01–0.001% in PCR and 0.01–0.0001% in NGS. The most problematic issue of molecular MRD is its limited applicability in children, i.e., major fusions (e.g., *RUNX1*::*RUNX1T1*, *CBFB*::*MYH11*, *PML*::*RARA*, *KMT2A*::*MLLT3*) are found in less than 40% of children with AML and major gene mutations found in adults (e.g., NPM1c, *FLT3*-ITD) are far less prevalent in children. In addition, it is well recognized that *RUNX1*::*RUNX1T1* and *CBFB*::*MYH11* transcripts may persist in the patient’s bone marrow while in long-term remission, and because of this, false-positive results may come out. The St. Jude Children’s Research Hospital (SJCRH) study in the U.S., which compared flow-cytometric MRD and molecular MRD, showed discrepant results (only 9.6% of the PCR-positive samples were flow-positive) and that PCR-MRD results did not have a prognostic impact when flow-MRD was negative [49]. The ELN MRD Working Party suggests a failure to reach a 3-log to 4-log reduction between the sample at diagnosis and at the end of treatment in adult patients with CBF-AML would be a relevant marker for subsequent relapse, but caution is needed whether it could be applied to children as well [50]. As a consequence, multiparametric flow cytometry (MFC)-MRD is considered to be the more preferred method for children with AML because of its wide applicability (>95% of the patients), although sensitivity is potentially lower (0.1–0.01%) than molecular MRD. Currently, there are two MFC-MRD approaches to target leukemia cells, namely, the leukemia-associated immunophenotype (LAIP) approach used in many study groups and the different-from-normal (DfN) aberrant immunophenotype approach used in the COG studies [51,52,53,54,55,56,57]. The LAIP approach is more complicated because it needs to select a patient-specific antigen combination. In contrast, the DfN approach employs a standardized panel which could potentially be applied to all patients regardless of the leukemia blast immunophenotype at diagnosis (i.e., does not require access to the diagnostic specimen) and has the strength that the method does not rely on the stability of a diagnostic LAIP during treatment, and therefore, the blasts can be detected even if an immunophenotypic shift occurs. Whichever approaches are applied, positive MRD at the end of one or two courses of induction therapies is shown to be the strongest predictor of poor outcomes in every previously reported clinical trial. Notably, the SJCRH AML02 study using the LAIP approach and the COG AAML0531 study using the DfN approach both showed the limited impact of morphological remission status on the negative MRD condition at the end of initial induction therapy [49,58]. 

### 2.3. Risk Stratification in Pediatric AML

Current risk stratifications used in most of the pediatric AML studies are based on combinations of leukemia-specific cytogenetic/molecular genetic abnormalities and MRD-based treatment response (Table 1). The NOPHO-DBH AML 2012 study (NCT01828489) by the Nordic Society of Paediatric Haematology and Oncology (NOPHO), Belgium, the Netherlands, and others are quite unique in that their risk stratification is strongly based on treatment response. Pediatric AML risk stratification has been focusing on determining the high-risk subsets of the patients assigned to receive HSCT in the first CR. Importantly, one should note whether HSCT truly improves the outcome of the high-risk patients. The outcome of a certain subset of the patients (e.g., *FUS*::*ERG*) has not improved by simply allocating the patients to receive HSCT and novel therapeutic strategies are urgently needed for these patients [59]. In the current genomic era, future risk stratification should focus more on identifying targetable lesions to incorporate molecularly targeted therapies. Success for APL using an introduction of All-Trans-Retinoic Acid (ATRA)-combined chemotherapy, and more recently of ATRA/arsenic trioxide combination therapy, is an ideal model, and many groups are starting to take this approach, as is the case with *FLT3*-ITD. 

## 3. Current Standard Therapy for Pediatric AML

### 3.1. Chemotherapy

Multi-agent combination chemotherapy is still a mainstay of treating children with AML. Key drugs are cytarabine and anthracyclines. Similar to adult AML, standard initial induction chemotherapy in children is based on the “3 + 7” regimen (seven days of low-to-intermediate dose cytarabine [LDAC] concurrent with three days of anthracyclines), but a third drug (e.g., etoposide) is often combined although its role is not fully established (Figure 2). The Japanese group introduced a unique prolonged schedule of induction therapy “ECM” (12 days in total) in the ANLL91 study in the early 1990s, on the basis of a high proportion of FAB-M4/M5 subtypes in children and frequent high leukocyte presentation and expected high efficacy of etoposide against monocytic AML [60]. Since then, this regimen has been used in the Japanese trials (AML99, AML-05, AML-12, and the currently ongoing AML-20), and was recently adopted in the NOPHO-DBH AML 2012 study as well [6,7,8,61,62]. Because of the significant prognostic impact of induction therapies and the limited number of pediatric AML patients, randomized questions to improve the outcome of children with AML have been mainly set at induction phases in the past pediatric AML studies worldwide. However, most of the study questions raised in the past trials have failed to show the impact on improved survival in pediatric AML (Table 2), specifically, the role of high-dose cytarabine (HDAC) [8,11,63,64,65], use of different types of anthracyclines [13,47,66], and addition of other cytotoxic drugs [10,12,67]. An exception is the addition of gemtuzumab ozogamicin (GO), an anti-CD33 antibody-drug conjugate (ADC). Cell surface antigen CD33 is expressed in more than 80% of the patients with AML, which makes this agent an attractive option. A randomized study, COG AAML0531, showed that the addition of GO 3 mg/m^2^ to ADE induction (and also to the second consolidation course) significantly improved the EFS (but not OS) of children with newly diagnosed AML [9]. Consequently, GO is currently approved by the U.S. Food and Drug Agency (FDA) for use in this setting and is regarded as a standard of care in the U.S. In the MyeChild01 study by the UK, Ireland, and France, a dose-finding study of GO (cohort 1, GO 3 mg/m^2^/dose on day 4 of initial induction consisting of cytarabine and mitoxantrone; cohort 2, GO on days 4 and 7; cohort 3, GO on days 1, 4, and 7) is being conducted (NCT02724163). 

Post-induction chemotherapies, namely, consolidation or intensification courses, are provided to all the patients who have achieved morphological CR to further consolidate the remission status. Drugs used are almost the same as induction chemotherapies, consisting mainly of cytarabine (generally includes HDAC) with or without anthracyclines and/or other additional drugs. Many of the questions regarding post-induction chemotherapies remain unsolved (Table 2), including the number of chemotherapy courses [6,47,70] and the addition of other cytotoxic drugs. Regarding the role of GO in post-induction therapy, COG AAML0531 showed the benefit of adding GO in a second consolidation course with HDAC and mitoxantrone (also in initial induction) [9]. However, the addition of GO (5 mg/m^2^/dose on days 1 and 21) at the very end of consolidation chemotherapies failed to improve both EFS and OS in the NOPHO-AML2004 study [71]. Currently, the Japan Children’s Cancer Group (JCCG) is evaluating the role of GO in post-induction phases by randomizing the intermediate-risk and high-risk patients to receive three additional courses of GO (3 mg/m^2^/dose)-combined or non-combined consolidation chemotherapy (jRCTs041210015) [62] 

Unlike the treatment of ALL, maintenance therapy is not a part of the standard of care for AML [72]. However, maintenance therapy using targeted drugs, including FLT3 inhibitors, might offer a benefit [73]. 

As mentioned previously, there is no clear evidence for the best anthracyclines of choice in AML chemotherapy. However, when comparing cumulative doses of different anthracyclines, the equivalence ratio is an issue, particularly in terms of late cardiotoxicity risks. For mitoxantrone and idarubicin, the ratio of 1:4–5 has been generally used for doxorubicin-equivalent doses. Recently, Feijen et al. reported a higher mean mitoxantrone conversion ratio of 10.5 (ratio of 0.5 for daunorubicin) based on cardiomyopathy risk assessment of the 28,423 childhood cancer survivors from the Childhood Cancer Survivor Study (CCSS), St. Jude Lifetime (SJLIFE), and Dutch Children’s Oncology Group (DCOG)-LATER study cohorts [69]. This data should be taken cautiously because the survivors included in this study were treated quite a long time ago (mostly between the 1960s and late 1990s) with various disease backgrounds (not only AML), and there have been no reports on increased cardiotoxicities from the UK or the Japanese group that had used mitoxantrone in their AML protocol since the late 1990s. The risk of late cardiotoxicity is something one should take into account for choosing kinds and doses of anthracyclines.

Because the principle AML therapy has been systemic therapy, the need for local therapy, including central nervous system (CNS)-directed therapy, is not clear. Most groups usually include intrathecal therapy (ITT) with cytarabine with or without methotrexate and corticosteroids in every chemotherapy course, but it is not evidence-based. In fact, adult AML studies generally do not include ITT. However, it is well recognized that children with AML (compared to adults) possess features with a higher risk of CNS disease and/or CNS relapse, such as higher leukocyte count at diagnosis and a higher prevalence of monocytic leukemia [74]. 

### 3.2. Hematopoietic Stem Cell Transplantation

The anti-leukemia effect of allogeneic HSCT relies on the cytotoxic effect of conditioning therapy and the immunological graft-versus-leukemia (GVL) effect by donor-derived cytotoxic immune cells; an approximately 60–70% DFS rate and 10–15% treatment-related mortality (TRM) is expected if transplanted in first or second CR [74]. Despite its potential risk of both acute and late toxicities, HSCT still plays an important role as a curative post-remission therapy for children with AML, although its indication is restricted to the high-risk subset (Table 1) [75]. Historically, both total body irradiation (TBI)-based and non-TBI-based (usually busulfan-based) myeloablative conditioning (MAC) were used in HSCT for children with AML. However, unlike ALL, both pediatric and adult studies (mainly retrospective studies) have shown similar or better results in favor of non-TBI-based conditioning for transplanting children with AML in terms of both efficacy and toxicity [76,77,78]. Therefore, intravenous busulfan (IV-BU)-based MAC is currently considered as standard; IV-BU in combination with melphalan and/or cyclophosphamide is generally used. Reduced-intensity conditioning (RIC) is an attractive option for children with AML, particularly in terms of reducing risks of late effects. Several retrospective analyses showed comparable outcomes between MAC and RIC, but an adult phase 3 randomized study for AML and MDS by the Blood and Marrow Transplant Clinical Trials Network (BMT CTN) demonstrated statistically significantly better relapse-free survival (RFS) in MAC [79,80,81]. BU-based MAC and RIC were randomly compared in children with AML in the MyeChild01 study, currently awaiting results. 

## 4. Novel Therapy for Pediatric AML

The outcomes of recently conducted pediatric AML studies are listed in Table 3. EFS ranges from 45% to 63% and OS from 65% to 80% [6,7,8,9,10,11,12,13,14,15,16,17]. As further improvements in outcomes for children with AML by conventional approaches are unlikely, the introduction of effective novel therapies to the current standard AML therapy would be a key solution. Several new classes of agents currently under development will be discussed.

### 4.1. FLT3 Inhibitors

FLT3 is a transmembrane ligand-activated receptor tyrosine kinase that is normally expressed by hematopoietic stem or progenitor cells and plays an important role in the early stages of both myeloid and lymphoid lineage development. An extracellular ligand binds and activates FLT3, promoting cell survival, proliferation, and differentiation through various signaling pathways, including PI3K, RAS, and STAT5. Mutations of the *FLT3* gene are found in approximately 30% of newly diagnosed adult AML cases (25% as ITDs and 10% as point mutations in the tyrosine kinase domain [TKD]) [38]. Frequencies in children with AML are lower, i.e., ITDs are found in 10% and TKD mutations in 6% of the cases [34]. Both *FLT3*-ITD and TKD mutations constitutively activate FLT3 kinase activity, resulting in proliferation and survival of AML. The presence of *FLT3*-ITD, not *FLT3*-TKD, is associated with poor outcomes both in children and adults with AML. FLT3 inhibitors are molecular-targeted agents that inhibit FLT3 signaling and are of two types. Type I inhibitors (midostaurin, gilteritinib) bind the FLT3 receptor in both the active and inactive conformational state of the FLT3 kinase domain, either near the activation loop or the ATP binding pocket, and are active against both ITD and TKD mutations. Type II inhibitors (sorafenib, quizartinib) bind specifically for the inactive conformation in a region adjacent to the ATP-binding domain. As a result of this binding affinity, type II FLT3 inhibitors prevent the activity of only ITD mutations but do not target TKD mutations. In terms of development history, midostaurin and sorafenib belong to the first generation, which was identified to have an FLT3 target among the various compounds with multi-targets. Quizartinib and gilteritinib belong to the second generation, which was originally designed to target FLT3, and therefore, more FLT3-specific compared to the first-generation inhibitors. Midostaurin is not active when used as monotherapy, but was developed for use in combination therapy [82]. Sorafenib, quizartinib, and gilteritinib all showed approximately 50% response rate for relapsed/refractory AML as monotherapy [83,84,85]. Each drug has specific toxicities, such as skin rash in sorafenib and QTcF prolongation in quizartinib for example. Current development of FLT3 inhibitors is focused on combination chemotherapy. In the QuANTUM-First trial, a randomized phase 3 study for newly diagnosed *FLT3*-ITD positive adult AML on a quizartinib combination, the addition of quizartinib significantly improved EFS and OS, and toxicities were comparable between the two arms [86]. Pediatric development of FLT3 inhibitors is lagging behind compared to adults. However, in the COG AAML1031 study, children with newly diagnosed high allelic ratio *FLT3*-ITD positive AML were eligible for receiving sorafenib combined therapy, and improved EFS was observed for the 72 children who took sorafenib compared to the 76 children who did not [71]. Table 4 shows the ongoing FLT3 inhibitor trials for children with AML. 

### 4.2. BCL2 Inhibitors

B-cell/CLL lymphoma-2 (BCL-2) family proteins regulate the intrinsic apoptosis pathway by integrating diverse pro-survival or pro-apoptotic intracellular signals. In AML, increased expression of BCL2 family proteins in leukemic blasts has been reported, and the majority of AML stem cells express aberrantly high levels of BCL2 and are dependent on BCL2 for survival. Furthermore, high expression of BCL2 has been associated with an inferior response to chemotherapy and poor survival among patients with AML. Venetoclax, a selective small-molecule BCL2 inhibitor, has been shown in preclinical studies to induce apoptosis in malignant cells that are dependent on BCL2 for survival. However, the single-agent venetoclax has had only modest activity in AML. Through downregulation of myeloid-cell leukemia 1 (MCL1) and induced expression of the pro-death proteins NOXA and PUMA, azacitidine or cytarabine synergistically inhibits the pro-survival proteins MCL1 and BCL-XL, thereby increasing the dependence of leukemia cells on BCL2. In fact, venetoclax combined with azacytidine or in combination with LDAC significantly prolonged the survival of adult patients with AML unfit for standard chemotherapy [87,88]. In the SJCRH phase 1 study for children with relapsed/refractory AML, venetoclax in combination with LDAC or HDAC was tested, and 360 mg/m^2^ venetoclax in combination with HDAC with or without idarubicin was determined to be the recommended phase 2 dose [89]. The overall response was 69%. Febrile neutropenia and invasive fungal infections were observed in 16% of the patients and one treatment-related death was observed. However, the treatment was tolerable overall. Currently, a phase 3 trial of venetoclax in combination with fludarabine, cytarabine, and GO for children with relapsed AML is ongoing (NCT05183035). This study is conducted as one of the sub-studies of PedAL/EuPAL initiatives, a global precision medicine master clinical trial that will test multiple targeted therapies simultaneously at various clinical sites, mainly in the U.S. and Europe. 

### 4.3. Menin Inhibitors

*KMT2A*-r AML accounts for 20–25% of pediatric AML. Recently, the updated retrospective study by the I-BFM was published; notably, nearly 50% of the patients failed the therapy even if their MRD was negative after the second induction [90]. Menin inhibitors are the most attractive class of agents for leukemia with *KMT2A*-r. Menin is a product of the *MEN1* tumor suppressor gene, which binds to the rearranged KMT2A complex and leads to the upregulation of leukemogenic genes (such as *HOX* and *MEIS1*), and thus to the subsequent development of acute leukemia. Menin inhibitors have shown selective, profound single-agent activity in *KMT2A*-r PDX models [91]. Menin inhibitors are potentially active against other subtypes of AML, such as NPM1c and *NUP98*-rearranged AML. NPM1c AML accounts for 20–30% of adult AML and 6% of pediatric AML and is generally associated with good prognosis. *NPM1*-coding protein nucleophosmin shuttles between the nucleus and cytoplasm during the cell cycle and is involved in diverse cellular processes, such as ribosome biogenesis, centrosome duplication, protein chaperoning, histone assembly, cell proliferation, and regulation of tumor suppressors TP53. However, mutated NPM1 (NPM1c) persists in the cytoplasm, and although the mechanism is not clear (but presumed to be a loss of function), NPM1c is associated with upregulation of *HOX* genes in a menin-dependent manner [92]. *NUP98*-rearranged AML accounts for less than 1% of adult AML and 7% of pediatric AML and is associated with unfavorable outcomes. It is known that NUP98 fusion proteins interact with KMT2A chromatin complexes and promote leukemogenesis. Inhibition of menin-KMT2A impairs leukemogenic gene expression and disrupts chromatin binding of menin, KMT2A, and NUP98 fusion proteins in mouse models [93]. Given the strong preclinical rationale justifying the use of menin inhibitors as a novel class of targeted therapy in acute leukemias, multiple clinical trials with these agents are in progress. The Syndax trial AUGMENT-101 is an industry-initiated first-in-human phase 1 clinical trial of the oral menin-inhibitor product revumenib (SNDX-5613) for both adults and children with relapsed/refractory acute leukemia with *KMT2A*-r or NPM1c [94]. Because revumenib is a substrate of cytochrome P450 3A4 (CYP3A4), two parallel dose-escalation cohorts, one without (Arm A) and one with (Arm B) strong CYP3A4 inhibitors, were conducted. There were no discontinuations or deaths due to treatment-related adverse events. Dose-limiting toxicities were asymptomatic grade 3 prolonged QTc in both arms. Notably, differentiation syndrome was observed in 16% of the patients. Overall response rate was 59%, and 73% of the patients achieving CR/CRh were MRD negative. Other menin inhibitor trials in children are also in progress.

### 4.4. Others

In addition to the cell signaling inhibitors against FLT3, BCL2, and menin, examples of novel therapies of interest for pediatric AML are immunotherapies (e.g., ADC, bispecific antibodies/T-cell engagers, chimeric antigen receptor T-cells [CAR-T]) targeting CD123 (expressed in nearly all AML subsets and leukemia stem cells), CD33, FLT3, or FOLR1 (targeting *CBFA2T3*::*GLIS2* fusion-positive AML), checkpoint inhibitors, cell-signaling inhibitors targeting MEK (NRAS and KRAS mutations are among the most common mutations in pediatric AML), and epigenetic modifiers (e.g., DNA methyltransferase inhibitors, histone deacetylase inhibitors, IDH1/IDH2 inhibitors) [95]. Finally, owing to the limited number of patients within each AML subgroup with a specific targetable disease, international cooperation (e.g., the PedAL/EuPAL initiatives) is crucial for effective drug development.

## 5. Conclusions and Future Directions

A quarter century of global efforts on clinical trials have contributed to improved outcomes for children with AML but are still tentative. Refinement in risk stratification based on leukemia biology and MRD, as well as the introduction of emerging novel therapies to contemporary therapy, through international collaboration, would be a key solution for further improvement in outcomes.

## Figures and Tables

**Figure 1 cancers-15-04171-f001:**
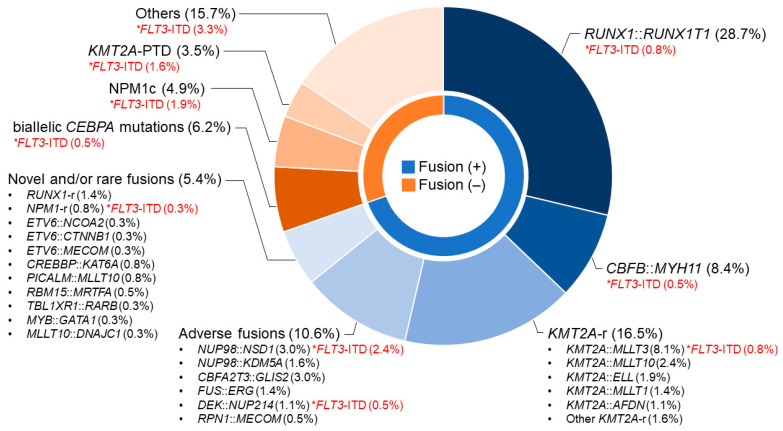
Genetic profiles of pediatric AML. Data is from the JPLSG AML-05 study. * Percentage of the patients with *FLT3*-ITD per total patients (e.g., patients with *RUNX1*::*RUNX1T1* and *FLT3*-ITD accounts for 0.8% of all 369 cases).

**Figure 2 cancers-15-04171-f002:**
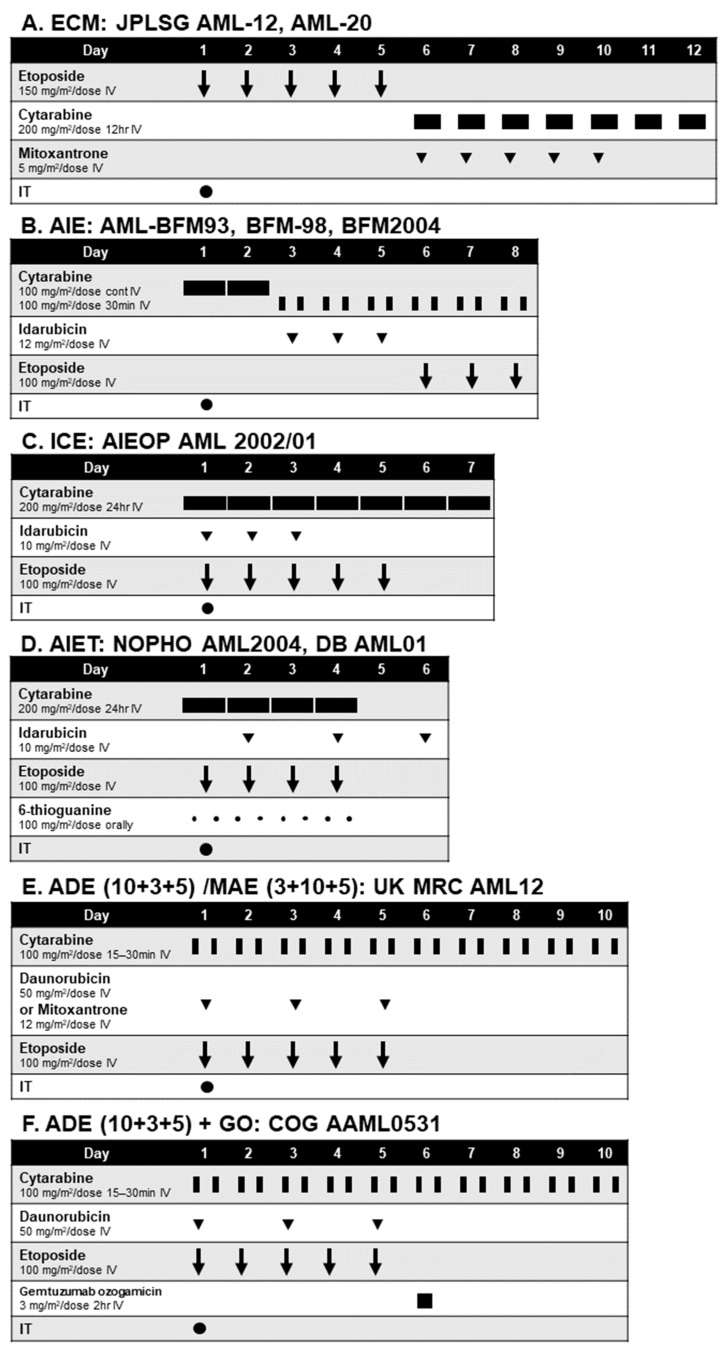
Induction therapy used in pediatric AML.

**Table 1 cancers-15-04171-t001:** Examples of risk stratification used in recent and ongoing pediatric AML studies.

	COG AAML1831	MyeChild 01	NOPHO-DBH AML 2012	JPLSG AML-20
SR	Low Risk 1 (LR1) CBF-AML - MRD@EOI1 < 0.05% - No *KIT* exon17 mutations - No other HR factors Mutated *NPM1*/*CEBPA*-bZip - MRD@EOI1 < 0.05% - No other HR factors Low risk 2 (LR2) Other than LR1 or HR	Standard risk (SR) Good-risk abnormalities * - MRD@EOI2 < 0.1% Intermediate-risk abnormalities ** - MRD@EOI1&2 < 0.1% Intermediate risk (IR) Good-risk abnormalities * - MRD@EOI2 > 0.1% Intermediate-risk abnormalities ** - MRD@EOI1 > 0.1% & EOI2 < 0.1% * Good-risk abnormalities CBF-AML Mutated *NPM1* (no *FLT3*-ITD) *CEBPA* double mutation (no *FLT3*-ITD) ** Intermediate-risk abnormalities t(9;11): *KMT2A*::*MLLT3* t(11;19): *KMT2A*::*MLLT1* Non-poor risk *KMT2A*-r Non-good/poor risk abnormalities	No high-risk (HR) factors MRD/BM blasts@EOI2 < 5% SR patients with inv(16)/t(16;16) receive a reduced number of consolidation courses	Low risk (LR) CBF-AML - No *FLT3*-ITD - MRD@EOI1 < 0.1% Intermediate risk (IR) CBF-AML & *FLT3*-ITD CBF-AML & MRD@EOI1 ≥ 0.1% Non-CBF-AML - No high-risk abnormalities † - MRD@EOI1 < 0.1%
HR	*FLT3*-ITD allelic ratio > 0.1 - No *NPM1/CEBPA*-bZip mutation *FLT3*-ITD allelic ratio > 0.1 - Mutated *NPM1/CEBPA*-bZip - MRD@EOI1 ≥ 0.05% Mutated non-ITD *FLT3* - MRD@EOI1 ≥ 0.05% RAM phenotype Unfavorable abnormalities: - inv(3)/t(3;3): *RPN1*::*MECOM* - t(3;21): *RUNX1*::*MECOM* - t(3;5): *NPM1*::*MLF1* - t(6;9): *DEK*::*NUP214* - t(8;16): *KAT6A*::*CREBBP* (≥90 days old) - t(16;21)(p11;q22): *FUS*::*ERG* - inv(16)(p13q24): *CBFA2T3*::*GLIS2* - t(4;11): *KMT2A*::*AFF1* - t(6;11): *KMT2A*::*AFDN* - t(10;11): *KMT2A*::*MLLT10* - t(10;11): *KMT2A*::*ABI1* - t(11;19): *KMT2A*::*MLLT1* - 11p15-r: any *NUP98* fusion - 12p13-r: any *ETV6* fusion - 12pdeletion: *ETV6* loss - −5/del(5q): *EGR1* loss - Monosomy 7 - 10p12.3-r: any *MLLT10* fusion No favorable/unfavorable abnormalities - MRD@EOI1 ≥ 0.05%	Intermediate-risk abnormalities ** - MRD@EOI2 > 0.1% Good-risk abnormalities * - MRD@EOC3 > 0.1% Poor-risk abnormalities - inv(3)/t(3;3)/abn(3q26) - −5/del(5q) - −7 - t(6;9): *DEK*::*NUP214* - t(9;22): *BCR*::*ABL1* - 12p abnormalities - t(4;11): *KMT2A*::*AFF1* - t(6;11): *KMT2A*::*AFDN* - t(10;11): *KMT2A*::*MLLT10* - t(5;11): *NUP98*::*NSD1* - t(7;12): *MNX1*::*ETV6* - inv(16)(p13q24): *CBFA2T3*::*GLIS2* - *FLT3*-ITD (no mutated NPM1, CBF) Other poor-risk categories Secondary leukemia without good-risk abnormalities Induction failure@EOI1	MRD/BM blasts@d22 of induction 1 ≥ 15% MRD@EOI2 ≥ 0.1–4.9% *FLT3*-ITD without mutated *NPM1*	Non-CR @EOI1 Non-CBF-AML - No high-risk abnormalities † - MRD@EOI1 ≥ 0.1% Non-CBF-AML - High-risk abnormalities † † High-risk abnormalities: Monosomy 7 −5/del(5q) inv(3)/t(3;3) *FLT3*-ITD (no CBF) t(9;22): *BCR*::*ABL1* t(4;11): *KMT2A*::*AFF1* t(6;11): *KMT2A*::*AFDN* t(10;11): *KMT2A*::*MLLT10* t(6;9): *DEK*::*NUP214* t(7;11): *NUP98*::*HOXA9* t(5;11): *NUP98*::*NSD1* t(11;12): *NUP98*::*KDM5A* inv(16)(p13q24): *CBFA2T3*::*GLIS2* t(16;21)(p11;q22): *FUS*::*ERG* t(7;12): *MNX1*::*ETV6* t(10;11): *PICALM*::*MLLT10* *TBL1XR1*::*RARB*

BM, bone marrow; CBF, core binding factor; EOI1/2, end of induction 1/2; EOC3, end of course 3; MRD, measurable residual disease. * Definition of Good-risk abnormalities in MyeChild01 study. ** Definition of Intermediate-risk abnormalities in MyeChild01 study. † Definition of High-risk abnormalities in JPLSG AML-20 study.

**Table 2 cancers-15-04171-t002:** Evidence for current standard therapy for pediatric AML.

Treatment Factors	Summary	Specific Data
Induction chemotherapy		
Cytarabine doses	Three randomized studies showed that there is not a clear impact of high-dose cytarabine in initial induction compared to low-dose or standard-dose cytarabine. High-dose cytarabine in the second induction may improve the outcome.	POG9421 [63] (*n* = 560): High-dose vs. standard-dose DAT in initial induction. No difference in CR and EFS. SJCRH AML02 [11] (*n* = 230): High-dose vs. low-dose ADE in initial induction. No difference in day 22 MRD, EFS, and OS. JPLSG AML-12 [8] (*n* = 324): High-dose vs. low-dose ECM in initial induction. No difference in end-of-induction MRD, EFS, and OS. Improved EFS for high-risk patients (*n* = 310) in AML-BFM93 by introducing HAM as a second induction [64]. Better RR, EFS, and OS with second induction HAM in t(8;21) patients (*n* = 78) in AML-BFM98 [65].
Anthracyclines	Overall, there is no clear evidence for the best anthracyclines of choice.	MRC AML12 [47] (*n* = 504): MAE vs. ADE. Use of mitoxantrone showed decreased RR and improved DFS over daunorubicin use, but no difference in EFS and OS. AML-BFM93 [66] (*n* = 358): AIE vs. ADE. Better day 15 bone marrow blast reduction with idarubicin compared to daunorubicin, but no difference in EFS and DFS. AML-BFM2004 [13] (*n* = 521): ADxE (liposomal daunorubicin) vs. AIE (idarubicin). No difference in RR, EFS, and OS.
Addition of other cytotoxic drugs	No clear evidence of adding cytotoxic drugs to cytarabine/anthracycline induction. However, one randomized study showed the benefit of adding GO to initial induction and second consolidation courses. Clofarabine may spare the use of anthracyclines and etoposide. Some groups use fludarabine to enhance the effect of cytarabine (FLA).	MRC-AML10 [67] (*n* = 359): DAT (6-thioguanine) vs. ADE (etoposide). No difference in CR, RR, DFS, and OS. COG AAML1031 [10] (*n* = 1097): Randomization to add bortezomib to each standard chemotherapy course failed to improve EFS and OS. COG AAML0531 [9] (*n* = 1022): ADE + GO (3 mg/m^2^) vs. ADE. Improved EFS (but not OS) and reduced RR in GO arm. SJCRH AML08 [12] (*n* = 262): Clofarabine + HDAC vs. high-dose ADE. No difference in EFS and OS. DB-AML-01 [16] (*n* = 112): Patients with t(8;21) or day 15 marrow blasts ≥ 5% received FLA + liposomal daunorubicin as second induction.
Post-induction chemotherapy		
Number of courses	A number of chemotherapy courses range from 4 to 6 (including induction) in recently conducted pediatric AML studies. Two retrospective analyses show benefit of an additional chemotherapy course for a subset of LR patients.	MRC-AML12 [47] (*n* = 270): 4 vs. 5 courses. No survival benefit for a 5th course of chemotherapy. Combined analysis of COG AAML0531 and AAML1031 studies [68] (*n* = 923) showed higher RR and lower DFS (but not OS) in a subset of LR patients who received 4 courses compared to those who received 5 courses. In the JPLSG AML-05 study [6] (*n* = 154), a reduction to 5 from 6 courses in the AML99 study (*n* = 89) resulted in increased RR in CBF-AML patients.
Addition of other cytotoxic drugs	No clear evidence of adding cytotoxic drugs to cytarabine/anthracycline chemotherapy. However, one randomized study showed the benefit of adding GO to initial induction and second consolidation courses.	COG AAML0531 [9] (*n* = 1022): MA + GO (3 mg/m^2^) vs. MA (second consolidation course). Improved EFS (but not OS) and reduced RR in GO arm. NOPHO-AML2004 [69] (*n* = 120): Addition of GO (5 mg/m^2^/dose on days 1 and 21) vs. no further therapy following the end of consolidation chemotherapies. No improvement in EFS and OS. COG AAML1031 [10] (*n* = 1097): Randomization to add bortezomib to each standard chemotherapy course failed to improve EFS and OS.
Maintenance therapy	No clear role of maintenance therapy. Major study groups no longer use maintenance therapy.	LAME89/91 [70] (*n* = 268): Maintenance therapy was introduced in LAME89 and randomized to receive or not receive maintenance in LAME91. No difference in EFS and OS.
Central nervous system-directed therapy		
CNS-directed therapy	Most groups usually include intrathecal therapy (ITT) in every chemotherapy course, but it is not evidence-based.	Previous AML-BFM studies included prophylactic CNS irradiation, due to the BFM-AML87 study results that the patients without CNS irradiation showed an increase in marrow relapses (not CNS relapses) compared to irradiated patients, but stopped since 2009 [18].

Abbreviations: BFM, Berlin-Frankfurt-Münster; COG, Children’s Oncology Group; DB, Dutch-Belgian; LAME, Enfant Leucemie Aigue Myeloblastique; JPLSG, Japanese Pediatric Leukemia/Lymphoma Study Group; MRC, Medical Research Council; NOPHO, Nordic Society of Paediatric Haematology and Oncology; SJCRH, St. Jude Children’s Research Hospital; ADE, cytarabine + daunorubicin + etoposide; ADxE, cytarabine + liposomal daunorubicin + etoposide; AIE, cytarabine + idarubicin + etoposide; AM, cytarabine + mitoxantrone; CNS, central nervous system; CR, complete remission rate; CBF, core-binding-factor; DAT, daunorubicin + cytarabine + 6-thioguanine; DFS, disease-free survival rate; EFS, event-free survival rate; FLA, fludarabine + cytarabine; GO, gemtuzumab ozogamicin; HAM, high-dose cytarabine + mitoxantrone; MAE, mitoxantrone + cytarabine + etoposide; MRD, measurable residual disease; OS, overall survival rate, RR, relapse rate.

**Table 3 cancers-15-04171-t003:** Comparison of recently completed pediatric AML studies.

Study (Years of Accrual)	No. of Patients	Risk Group/ Treatment Arm	Cumulative Anthracycline Doses				No. (%) of Patients Treated with CR1 HSCT	EFS, % OS, % (Years)	References
			Daunorubicin	Mitoxantrone	Idarubicin	Others			
JPLSG AML-05 (2006–2010)	443	LR	–	25	20	–	46 (10)	54 (3) 73 (3)	Tomizawa et al., 2013 [6] Hasegawa et al., 2020 [7]
		IR/HR	–	IR:55/HR40	IR:20/HR:10	–			
JPLSG AML-12 (2014–2018)	359	CBF SR	–	40	20	–	40 (11)	63.1 (3) 80.3 (3)	Tomizawa et al., 2018 [8]
		nCBF-SR/HR	–	nCBF-SR:55 HR:40	nCBF-SR:20 HR:10	–			
COG AAML0531 (2006–2010)	1022	No HSCT	300	48	–	–	157 (15)	53.1 (3) * 69.4 (3) *	Gamis et al., 2014 [9]
		HSCT	300	–	–	–			
COG AAML1031 (2011–2016)	1097	LR	300	48	–	–	85 (8)	45.9 (3) 65.4 (3)	Aplenc et al., 2020 [10]
		HR	300	–	–	–			
SJCRH AML02 (2002–2008)	230	No HSCT	300	20 * or 50	–	–	59 (26)	63.0 (3) 71.1 (3)	Rubnitz et al., 2010 [11]
		HSCT	300	–	–	–			
SJCRH AML08 (2008–2017)	262	HD-ADE	300	36	–	–	81 (31)	52.9 (3) ** 74.8 (3) **	Rubnitz et al., 2019 [12]
		Clo + Ara-C	150	36	–	–			
AML-BFM2004 (2004–2010)	611	ADxE	–	SR:20/HR:40	14	DNX: 240	NA	55 (5) 74 (5)	Creutzig et al., 2013 [13]
		AIE	–	SR:20/HR:40	50	–			
AIEOP AML2002/01 (2002–2011)	482	–	–	50	60	–	141 (29)	55 (8) 68 (8)	Pession et al., 2013 [14]
NOPHO AML2004 (2004–2009)	151	–	–	30	48	–	22 (15)	57 (3) 69 (3)	Abrahamsson et al., 2011 [15]
DB-AML-01 (2010–2013)	112	AM	–	30	36	–	NA	52.6 (3) 74.0 (3)	De Moerloose et al., 2019 [16]
		FLA-DNX	–	–	36	DNX: 180			
ELAM02 (2005–2011)	438	SR	80	60	–	AMSA: 300	119 (27)	57 (4) 73 (4)	Petit et al., 2018 [17]
		IR/HR	–	60	–	AMSA: 300			

Abbreviations: AIEOP, Associazione Italiana di Ematologia e Oncologia Pediatrica; BFM, Berlin-Frankfurt-Münster; COG, Children’s Oncology Group; DB, Dutch-Belgian; ELAM, Enfant Leucemie Aigue Myeloblastique; JPLSG, Japanese Pediatric Leukemia/Lymphoma Study Group; NOPHO, Nordic Society of Paediatric Haematology and Oncology; SJCRH, St. Jude Children’s Research Hospital; AM, cytarabine + mitoxantrone; AMSA, amsacrine; Ara-C, cytarabine; Clo, clofarabine; CR1, first complete remission; CBF, core-binding-factor; DNX, daunoxome; EFS, event-free survival rate; FLA-DNX, fludarabine + cytarabine + daunoxome; HR, high-risk; HSCT, hematopoietic stem cell transplantation; IR, intermediate-risk; LR, low-risk; NA, not available; OS, overall survival rate, SR, standard-risk. * GO arm. ** Clo + Ara-C arm.

**Table 4 cancers-15-04171-t004:** Going FLT3 inhibitor trials in children with AML.

Trial (ClinicalTrials.gov Identifier)	Regimen	Key Eligibility	Phase (No. Patients)	Current Status
Novartis (NCT03591510)	Midostaurin + chemo	Children (3 mo–17 yo) *FLT3*-mutated AML	Phase 2 (*n* = 23)	Recruiting 33 sites: US, Austria, Czechia, Germany, Greece, Italy, Poland, Russia, Slovenia, Turkey, Jordan, Japan, Korea
COG AAML1831 (NCT04293562)	Gilteritinib + chemo	Children (2 yo–21 yo) *FLT3*-ITD (AR > 0.1)+ AML *FLT3*-TKD + AML	Phase 3	Recruiting
Astellas (NCT04240002)	Gilteritinib + chemo	Children, AYA (6 mo–21 yo) r/r *FLT3*-ITD + AML	Phase 1/2 (*n* = 97)	Recruiting 19 sites: US, Canada, Germany, Italy, Spain, UK
Daiichi Sankyo/ITCC/COG (NCT03793478)	Quizartinib + chemo	Children, AYA (1 mo–21 yo) r/r *FLT3*-ITD + AML	Phase 1/2 (*n* = 65)	Recruiting 36 sites: US, Canada, Belgium, Denmark, France, Italy, Netherlands, Spain, Sweden, UK, Israel

Abbreviations: COG, Children’s Oncology Group; ITCC, Innovative Therapy for Childhood Cancer; AR, allelic ratio; AYA, adolescents and young adults; *FLT3*-ITD, internal tandem duplication of *FLT3* gene; *FLT3*-TKD, tyrosine kinase domain mutation of *FLT3* gene; r/r, relapsed/refractory.

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
