# Peer review of "Risk-Stratified Therapy for Pediatric Acute Myeloid Leukemia"

_cancers, 2023, doi:10.3390/cancers15164171_

Round 1

Reviewer 1 Report

The topic is not new, but of utmost importance in our field, since major international efforts were conducted for decades to improve the survival in children with AML. The authors clearly have a great knowledge on a chosen topic and are presenting their own results (regarding genetic background and therapeutical approaches), which are excellent, but the manuscript has two main flaws: overall composition of the article is a bit chaotic and English need to be majorly corrected.

Regarding first comment, please, choose one or two topics of major interest and focus on it - conclusion is very general and not directed to the main text. The manuscript is offering a good literature review and important authors` data, but the way it is all presented need to be substantially corrected.  

Already mentioned above. 

Author Response

To Reviewer #1:

Thank you for your variable comments on our manuscript entitled “Pediatric acute myeloid leukemia,” now the title has changed to “Risk-stratified therapy for pediatric acute myeloid leukemia.”

The manuscript was condensed and revised to conform as closely as possible to your suggestions. The following revisions were made.

Regarding the second comment, the whole revised manuscript went through an English editing process by the native English speaker.

For the first comment, when we were invited to submit this review article to the journal, what the editors asked us to write was a comprehensive review of pediatric de novo AML. That is why the contents of the previous version of the manuscript was comprehensive rather than focusing on limited topics. However, we do agree with the point you had raised. Therefore, to establish both demands, we decided to focus on risk-stratified therapy for pediatric de novo AML and changed the title accordingly. Regarding the main text, we suppose the descriptions on standard chemotherapy was lengthy which made the initial version of the draft chaotic. Therefore, we eliminated most of these parts and made a new Table 2 instead to summarize them. We hope that these revisions are satisfactory.

Reviewer 2 Report

Well written review with all the basic information in pediatric acute myeloid leukemia but limited originality. Please add specific criteria for a decision to proceed in hematopoietic stem cell transplantation.

No comment

Author Response

To Reviewer #2:

Thank you for your variable comments on our manuscript entitled “Pediatric acute myeloid leukemia,” now the title has changed to “Risk-stratified therapy for pediatric acute myeloid leukemia.”

The manuscript was condensed and revised to conform as closely as possible to your suggestions. The following revisions were made.

When we were invited to submit this review article to the journal, what the editors asked us was to write a comprehensive review of pediatric de novo AML. That is why the originality is limited, however, we decided to focus on risk-stratified therapy for pediatric de novo AML and changed the title accordingly. Regarding the main text, we eliminated most of the descriptions on standard chemotherapy part which was lengthy, and made a new Table 2 instead to summarize them. We hope that these revisions are satisfactory.

P.14 lines 14-15: Regarding the indication of stem cell transplantation, the sentence “its indication is restricted to the high-risk subset (Table 1)” is clearly stated, and the definition of “high-risk” used in the several major pediatric AML cooperative study groups are clearly listed in Table 1.

Reviewer 3 Report

The authors summarized the current knowledge on mutational landscape and on therapeutic option in pediatric acute myeloid leukemia. The paper is interesting, contains many useful information and comparison with the adult counterpart. 

There are just few point to clarify:

1.     The authors stated that Hispanic and American African children have worse prognosis compared to Caucasian. How about Asian children? There are differences in incidence and prognosis? 

2.     Regarding leukemia specific cytogenetic/molecular alterations they reported the genetic profile of a Japanese series including 369 patients (fig1). Does this profile coincide with genetic profile of children from different ethnicities?

3.     In the section 2-3 table 1 is missing.

4.     In the same section current criteria to stem cell transplantation should be specified

5.     In the section 3-1-3 and 3-2-2 discussing the addition of drug other that cytarabine and anthracycline to induction or post remission therapy a comment on fludarabine should be added.

6.     A short section for therapies in relapsed disease would be interesting

7.     The section 3-4 on the most novel option deserve to be expanded.   

Author Response

To Reviewer #3:

Thank you for your variable comments on our manuscript entitled “Pediatric acute myeloid leukemia,” now the title has changed to “Risk-stratified therapy for pediatric acute myeloid leukemia.”

The manuscript was condensed and revised to conform as closely as possible to your suggestions. The following revisions were made.

The authors summarized the current knowledge on mutational landscape and on therapeutic option in pediatric acute myeloid leukemia. The paper is interesting, contains many useful information and comparison with the adult counterpart.

There are just few point to clarify:

  1. The authors stated that Hispanic and American African children have worse prognosis compared to Caucasian. How about Asian children? There are differences in incidence and prognosis?

P.5 lines 2-6: We added a sentence regarding this issue as follows;

There is no data that directly analyzed differences between Asian children and other ethnicities; however, the literature shows higher prevalence of t(8;21) (RUNX1::RUNX1T1)-positive AML in Asian populations (approximately 30% compared to 12%–14% of the U.S. or European patients).

  1. Regarding leukemia specific cytogenetic/molecular alterations they reported the genetic profile of a Japanese series including 369 patients (fig1). Does this profile coincide with genetic profile of children from different ethnicities?

P.5 lines 13-15: We added a sentence regarding this issue as follows;

Distribution of genetic profiles is similar to the other groups in the U.S. or Europe except that the proportion of patients with RUNX1::RUNX1T1 is high in the Japanese cohort, as mentioned previously.

  1. In the section 2-3 table 1 is missing.

I am very sorry about this, but I am not sure why this had happened. Table 1 illustrates the risk group definition of several ongoing major pediatric AML studies (COG AAML1831, MyeChild01, NOPHO-DBH AML 2012, and JPLSG-AML-20). I will do my best to properly upload the Tables when submitting the revised manuscript.

  1. In the same section current criteria to stem cell transplantation should be specified

P.14 lines 14-15: Regarding the indication of stem cell transplantation, the sentence “its indication is restricted to the high-risk subset (Table 1)” is clearly stated, and the definition of “high-risk” used in the several major pediatric AML cooperative study groups are clearly listed in Table 1.

  1. In the section 3-1-3 and 3-2-2 discussing the addition of drug other that cytarabine and anthracycline to induction or post remission therapy a comment on fludarabine should be added.

Table 2: Because we decided to focus on risk-stratified therapy for pediatric de novo AML and changed the composition of the main text accordingly, we eliminated most of the descriptions on standard chemotherapy part which was lengthy, and made a new Table 2 instead to summarize them. We, therefore, added a comment on fludarabine in Table 2 as well.

  1. A short section for therapies in relapsed disease would be interesting

Thank you very much. We do agree with your comment. However, other reviewer had asked us to focus on fewer topics, and that is why we decided to change the manuscript to focus on risk-stratified therapy for pediatric de novo AML.

  1. The section 3-4 on the most novel option deserve to be expanded.

Thank you very much. We do agree with your comment. However, other reviewer had asked us to focus on fewer topics, and that is why we decided to change the manuscript to focus on risk-stratified therapy for pediatric de novo AML.

Round 2

Reviewer 1 Report

Authors made major improvements in the manuscript - it is greatly shaped now, information are comprehensive, put in logical order and very educative, as for junior, as well for senior colleagues. Thank you for putting extra efforts, which resulted in excellent article.